# The Application of Manual Techniques in Masticatory Muscles Relaxation as Adjunctive Therapy in the Treatment of Temporomandibular Joint Disorders

**DOI:** 10.3390/ijerph182412970

**Published:** 2021-12-08

**Authors:** Piotr Urbański, Bartosz Trybulec, Małgorzata Pihut

**Affiliations:** 1Department of Physiotherapy, Faculty of Health Sciences, Jagiellonian University Medical College, 9 Medyczna Str., 30-688 Krakow, Poland; bartosz.trybulec@uj.edu.pl; 2Institute of Physical Culture, PWSZ, 2 Kościuszki Str., 33-300 Nowy Sacz, Poland; 3Department of Prosthetic Dentistry, Faculty of Medicine, Jagiellonian University Medical College, 4 Montelupich Str., 31-155 Krakow, Poland; malgorzata.pihut@uj.edu.pl

**Keywords:** temporomandibular disorder, TMD, physiotherapy, rehabilitation, manual therapy, post-isometric relaxation, myofascial release

## Abstract

Temporomandibular disorders (TMD) are primarily characterized by pain as well as disorders concerning the proper functioning of individual elements of the stomatognathic system (SS). The aim of the study was to compare the degree of relaxation of the anterior part of the temporal muscles and the masseter muscles, achieved through the use of post-isometric relaxation and myofascial release methods in patients requiring prosthetic treatment due to temporomandibular joint disorders with a dominant muscular component. Sixty patients who met the inclusion criteria were alternately assigned to one of the two study groups, either group I—patients received post-isometric relaxation treatment (PIR), or group II—patients received myofascial release treatment (MR). The series of ten treatments were performed in both groups. The comparative assessment was based on physiotherapeutic examination, a surface electromyography (sEMG) of the anterior temporal and masseter muscles and the intensity of spontaneous masticatory muscle pain, assessed using the Visual Analogue Scale (VAS). We observed a significant decrease in the electrical activity of examined muscles and a significant drop in the intensity of spontaneous pain in the masticatory muscles both in group I and II. There were no significant differences between groups. Both therapeutic methods may be used as successful forms of adjunctive therapy in the prosthetic treatment of TMD. The trial was registered with an international clinical trials register.

## 1. Introduction

Temporomandibular disorders (TMD) have recently become one of the most common dental diseases, and at the same time constitute a serious social problem. Epidemiological data indicate that approximately 40–60% of the population has at least one symptom of stomatognathic system (SS) dysfunction [1]. This high prevalence is related to its multidimensional etiology [1,2,3,4,5], diagnostic and therapeutic difficulties [6], and the lack of knowledge even among dentist on TMD management [7].TMD are primarily characterized by pain in the pre-aural area as well as disorders concerning the proper functioning of particular elements of the SS. These symptoms are often accompanied by hypertonia, hypertrophy and asymmetry in the functioning of masticatory muscles.

Effective treatment of such disorders depends on, among other things, an elimination of causes responsible for SS malfunction as well as a direct intervention within the muscular system in order to restore its physiological tension and reduce pain [1]. Clinicians and researchers also emphasize the importance of providing conservative treatment first, before introducing irreversible methods [1,3,8,9,10]. Therefore, currently recommended approaches in TMD therapy include primarily prosthetic and orthodontic treatment [1,8,10], pharmacotherapy [1,11,12], as well as rehabilitation and education as forms of supportive treatment [1,2,3,13,14,15,16,17,18]. One of the most commonly used forms of conservative treatment is physiotherapy, including manual therapy [19,20,21,22,23,24,25,26]. The use of these interventions with simultaneous prosthetic treatment significantly accelerates the restoration of proper functioning of the masticatory organ [1,13,14]. The most preferred methods of manual therapy are relaxation techniques, the use of which does not cause pain sensations, such as post-isometric muscle relaxation (PIR) and myofascial release (MR). The PIR method, based mainly on neurophysiological inhibitory (relaxing) mechanisms, allows for reduced muscle tension immediately after their isometric contraction and minimizes pain sensations related to the muscular system [15,27]. The MR method, based on the so-called phenomenon of relaxation, removes restrictions within the myofascial system, thus reducing pain and muscles tension [28,29,30,31].

The comparative studies on the effectiveness of PIR and MR applied in TMD patients have not been conducted so far. There are only individual scientific reports on assessing these methods separately in the treatment of masticatory organ. The reports indicate the effectiveness of PIR in increasing the range of mandibular motion in patients treated due to the presence of trigger points and an increased tension of the masseter muscles [16]. On the other hand, the results of studies assessing the clinical effects of MR in the treatment of such disorder show its effectiveness in reducing pain and improving the range of movement of the mandible [17]. Despite the limited data on the use of these methods in the context of the masticatory system, there are numerous studies assessing the effects of PIR [27,32,33,34,35] and MR [36,37,38,39] in the therapy of muscles located within the trunk and limbs. The reports indicate that PIR and MR are highly effective in improving the functioning of the musculoskeletal system, increasing the range of motion in the joints, alleviating pain and decreasing muscle stiffness.

The aim of the study was to compare the degree of relaxation of the anterior part of the temporal muscles and the masseter muscles, achieved with the use of post-isometric relaxation and myofascial release methods in patients requiring prosthetic treatment due to temporomandibular disorders with a dominant muscular component.

## 2. Materials and Methods

The study material consisted of 60 patients of both sexes, aged 19 to 40, who had been referred to the Dental Prosthetic Clinic of the Institute of Dentistry, Jagiellonian University, in Krakow, Poland, for the prosthetic treatment of pain-related TMD, accompanied by increased masticatory muscles tension, and met the inclusion criteria. All patients were informed about the terms of participation in the study and provided written, informed consent during enrolment. The research was carried out in accordance with the guidelines of the Helsinki Declaration and obtained the consent of the Bioethics Committee of the Jagiellonian University (No 122.6120.129.2016 dated 25 May 2016). The trial was registered with clinicaltrials.gov (No NCT05091996 accessed on 25 October 2021).

The following inclusion criteria were applied: (i) patients with clinical features fulfilling the Axis I, Group Ia of RDC/TMD [40] with local or myofascial pain and increased tension of masticatory muscles, (ii) absence of temporomandibular disc displacement with or without reduction, (iii) spontaneous pain in the masticatory muscles, for the duration of at least one month before treatment, (iv) good general health (absence of chronic diseases which may affect temporomandibular joint or the masticatory muscles), (v) full dental arches with natural teeth or missing teeth replaced with fixed dental prostheses and (vi) written consent to participate in the study.

The exclusion criteria included the following: (i) occurrence of a face or head injury during participation in the research, (ii) open wounds in the area where the therapy was carried out, (iii) sudden illness of the patient preventing participation in the study, and/or (iv) the will to terminate participation in the study.

Patients who fulfilled the inclusion criteria were alternately assigned to one of the two study groups. In group I, which comprised of 30 patients (22 women and 8 men; 28 ± 5.31 years) PIR treatments were performed, and in group II, which also consisted of 30 patients (25 women and 5 men; 28 ± 5.10 years) MR treatments were performed. Each patient received 10 treatment sessions for 10 consecutive days except Sundays.

In both groups, a clinical examination was performed in the following order:(1)General dental examination, based on the Diagnostic Criteria of Temporomandibular Disorders (RDC/TMD) [40];(2)Physiotherapeutic examination of the masticatory organ;(3)Electromyographic examination (sEMG) of the anterior part of the temporal and masseter muscles;(4)Assessment of the intensity of muscle pain using the VAS (visual analogue scale).

All parts of the clinical examination were performed by the same experienced physiotherapist three times, firstly immediately before commencing treatment (measurement 1), secondly after the last of 10 relaxation treatments (measurement 2), and lastly on the 4th day after the end of therapy (measurement 3). The dental examination was an exception, as it was performed once by a doctor qualifying patients during enrolment. All clinical examinations and treatment sessions occurred prior to the introduction of occlusal splint therapy, which was used as a primary treatment of TMD. Thus, both PIR and MR methods (used as supportive therapies) were evaluated individually, with no impact of the primary treatment on the patients’ condition.

### 2.1. Examination

#### 2.1.1. Physiotherapeutic Examination

The physiotherapeutic examination was performed in compliance with the guidelines of the Manual Functional Analysis (MFA) of the masticatory apparatus according to Bumann [3,41].

#### 2.1.2. Electromyographic Examination of the Masticatory Muscles

The electrical activity of the anterior part of the temporal and masseter muscles was assessed in accordance with the assumptions of surface electromyography (sEMG) [42,43,44]. The study was performed using a two-channel device—NeuroTrac MyoPlus 4 (Verity Medical Ltd., Romsey, Hampshire, United Kingdom) and surface electrodes (Noraxon Inc., Scottsdale, Arizona, USA). The examinations of the same muscles on the right and left sides was performed simultaneously. The anterior part of the temporal muscles was assessed first, followed by the masseter muscles. The mean value of the thirty-second sEMG measurement was selected as a parameter to describe the degree of electrical activity of the muscle [42].

During the examination, the patient was in a sitting position, without back support, with legs on the ground and palms on thighs. The sEMG test was conducted in the rest position of the mandible in order to standardize the starting position for the measurement. The respondent was instructed in advance to maintain their natural muscle tension during the test. It should be noted that maintaining the increased muscle tension if the examined person resulted in mandible-position change, thus bringing the dental arches into contact.

Silver-chloride electrodes, with a constant distance between the poles (20 mm), were used for the tests. They were placed parallel to the muscle fibers, after the skin was degreased with alcohol. Two measuring electrodes were used to assess the electrical activity of one muscle and a reference electrode was placed on the forehead (Figure 1 and Figure 2). In order to ensure the repeatability of the electrode positions, the relationships of selected anthropometric points were used and a photographic documentation of the sEMG test was created each time. On the temporalis muscle, electrodes were placed vertically in the anterior part of temporal fossa, posterior to the zygomatic process of the frontal bone, in the space between the zygomatic arch and the inferior temporal line (Figure 1). On the masseter muscle, electrodes were located in the space between the gonial angle and the inferior border of the zygomatic bone (Figure 2).

#### 2.1.3. Assessment of the Intensity of Spontaneous Pain in the Masticatory Muscles

The intensity of spontaneous pain in the masticatory muscles was assessed based on a 10-point VAS. This scale included 11 pain grades from 0 to 10, where 0 indicated no pain at all, 5 indicated moderate pain, and 10 represented the strongest pain imaginable.

### 2.2. Interventions

#### 2.2.1. Post-Isometric Muscle Relaxation Treatments

During PIR procedures, the patient was placed in a supine position, with the head placed neutrally. Due to the fact that the muscles lifting the mandible also contract during lateral movements of the mandible, the procedure was performed in two stages. First, the muscles were stretched in the direction of the abduction, then in the directions of the lateral movements of the mandible. The series of treatments in the first and second stages were repeated six times during one visit.

##### Relaxing the Mandibular Adductors

When performing this technique, the therapist placed his thumbs on the patients’ premolar and molar chewing surfaces and then passively abducted the mandible, until the so-called functional barrier was achieved. In this position, the patient performed an isometric contraction of the mandibular adductors using around 20% of their maximum force—the contraction initiated was balanced by the therapist’s hands (Figure 3A). After 10 s of isometric tension, the patient relaxed their muscles, and the therapist abducted the mandible to reach a new functional barrier (Figure 3B). During one treatment, the described cycle was repeated three times, starting from the previously obtained movement barrier each time [15].

##### Relaxing the Muscles Responsible for Lateral Movements of the Mandible

The relation between the mandible and the maxilla in the starting position for this procedure was the same as in the mandibular physiologic rest position. First, the therapist placed one hand so that the thenar, thumb, and index finger embraced the mandibular body, angle, and ramus. At the same time, he stabilized the patient’s head with the other hand, positioning it on the temple on the opposite side. (Figure 4A). Then, he carried out passive lateral translation of the mandible until the first tissue counter-tension occurred. In this position, the patient performed isometric muscle contraction using about 20% of maximum force, as to initiate the movement towards the starting position. At the same time, the therapist stabilized the head and mandible, and balanced the contraction generated by the subject. After 10 s of isometric tension, the patient relaxed his muscles. Then, the therapist passively deepened the lateral movement of the mandible until a new functional barrier was reached (Figure 4B). During one treatment, the described cycle was repeated three times, both to the right and left side, each time starting from the previously obtained movement barrier [15].

#### 2.2.2. Myofascial Release Treatment

The MR procedure was performed successively in the area of the anterior parts of the temporal muscles (Figure 5), the superficial parts of the masseter muscles (Figure 6) and the sternocleidomastoid muscles (Figure 7). During one session, the above cycle was repeated separately on both sides six times.

Soft tissue mobilization started from the area of proximal attachment of each muscle. The patient was lying on his back, with his head turned to the side. The therapist, using the pad of the first finger, eliminated the tissue slack by sliding tissues in the caudal direction. During this movement, when the tissue barrier was reached, he maintained tension on the displaced structures. Then, he shifted them gently until the physiological tissue barrier was achieved. A single application of the MR procedure consisted of one shift movement of the soft tissues along the muscle undergoing therapy [28].

### 2.3. Statistical Analysis

A statistical analysis was performed using Statistica 12.0 (Statsoft, Tulsa, OK, USA). Continuous variables were expressed as a mean with standard deviation or a median for the first and third quartile (q1–q3). The normality of the data was assessed using the Shapiro–Wilk test. Differences between groups for normally distributed continuous variables were compared using the Student’s *t*-test, and for non-normally distributed continuous variables the Mann–Whitney U test was used. The Chi-square tests were used for categorical data. The ANOVA test for repeated measures was used to test the changes in the mean of electrical potentials using the Bonferroni post hoc test. The Friedmann test and the Wilcoxon test were used to examine the changes in the intensity of spontaneous pain in the masticatory muscles. All statistical analyses were performed using IBM Corp. released in 2019. IBM SPSS Statistics for Windows, Version 26.0. Armonk, NY, USA: IBM Corp. The significance level was assumed to be α = 0.05 for two-sided tests.

## 3. Results

Before the therapy there was found to be no difference between the groups in the distribution of sex, age, electromyographic tension of the anterior part of the temporal and masseter muscles, and the intensity of spontaneous pain in the masticatory muscles (Table 1).

### 3.1. Results of the Physiotherapeutic Examination of the Masticatory Apparatus in Both Groups

A comparison of the physical symptoms found during the first and the control examinations showed an improvement in the examined parameters in both groups (Table 2 and Table 3). The following effects were noticed:Improvement of the path and range of mandibular abduction movement both in group I and group II.Improvement of the range of lateral movements of the mandible in group I and group II.Reduction of the pain in the masticatory muscles during palpation in group I and group II.Reduction of pain in the temporomandibular joints during palpation in group I and group II.Reduction of pain in joints, periarticular structures or muscles during passive mandibular movements in group I and group II.

### 3.2. The Results of Electromyographic Examination of the Masticatory Muscles Obtained in Group I and Group II

The ANOVA tests revealed significant differences between the mean values of the electrical potentials obtained in the three electromyographic measurements of the anterior temporal and masseter muscles, both on the left and right side in group I (Table 4) and group II (Table 5). In both groups, the Bonferroni post hoc tests demonstrated a statistically significant difference in the mean values of the studied parameter between the 1st and 2nd measurement, and between the 1st and 3rd measurement for both the anterior part of the temporal muscles and the masseter muscles, on the right and left sides. A significant decrease in electrical activity of both muscles was observed, on the right and left, in both groups. However, there were no significant differences in the mean values of the electrical potentials of the examined muscles between measurements 2 and 3.

### 3.3. The Results of the Evaluation of the Intensity of Pain in the Masticatory Muscles in Group I and Group II

In both group I and group II, significantly higher values of the intensity of spontaneous pain in the masticatory muscles were declared by the participants of the study for measurement 1, compared to measurement 2 and measurement 3 (Table 6).

### 3.4. Comparison of Changes in the Electrical Activity of the Examined Muscles between Groups

The comparison of changes in the electrical activity of the anterior part of the temporal muscles during the first and control examinations between the groups showed no statistically significant differences. A greater decrease in the mean values of the electric potentials of the masseter muscles occurred in group II, but this relationship did not reach statistical significance, and were found to be, for the right side, *p* = 0.10, for the left side, *p* = 0.11 (Figure 8).

### 3.5. Comparison of Changes in the Intensity of Spontaneous Pain in the Masticatory Muscles between Group I and Group II

There were no significant differences in the change in the intensity of spontaneous pain in the masticatory muscles assessed during the first examination and follow-up examinations between patients in group I and patients in group II (Table 7).

## 4. Discussion

Current clinical guidelines for the treatment of TMD clearly indicate the need for conservative therapy, such as physiotherapy and manual therapy [1,3,10]. These interventions directly affect the muscular system, in order to reduce its excessive tension, and thus minimize pain in the area of the temporomandibular joints, and in muscles involved in the chewing process [1,3,19,20,21,22,23,24,25,26].

When analyzing the collected results, a significant decrease in the electrical activity of the anterior part of the temporal and masseter muscles was observed, as well as a significant decrease in the intensity of spontaneous pain in the masticatory muscles after a series of both PIR and MR treatments. There were also no significant differences in the improvement of the measured parameters between study groups.

To the authors’ knowledge, comparative studies evaluating the effects of the post-isometric muscle relaxation and myofascial release in patients treated for pain-related TMD, accompanied by increased muscle tension, have not been carried out as of yet. There are few scientific reports in the available literature assessing each of these methods separately in the context of the masticatory muscles. It is also noticeable that there are limited data on assessing manual therapy interventions with the use of sEMG in this field of medicine. Blanco et al. [16] conducted a study comparing the efficacy of PIR and positional release (strain/counterstrain) in the treatment of myofascial pain, manifested by latent myofascial trigger points (MTrPs) and located in the masseter muscles. A significant increase in the pain-free active mandibular range of motion was observed in subjects who underwent a single PIR procedure, while there was no significant change in the examined parameter in the group of subjects undergoing the positional release method. Ibanez-Garcia et al. [17] attempted to compare the therapeutic effects of the neuromuscular technique and the positional release method among patients diagnosed with MTrPs located within the masseter muscles. The neuromuscular technique described by the authors is a specific modification of MR intervention performed for the purposes of this research—the procedure was similar, but it was performed with the use of a lubricant. Nevertheless, both methods were found to be highly effective in reducing the pressure pain threshold (PPT) and increasing the maximum pain-free mandibular active range of motion, while a moderate effect on reducing the intensity of local pain induced by palpation was observed. Despite the fact that the measurement tools used in the cited studies differ from those used in our report, the results of these researches are consistent with each other [16,17]. Similar to the present study, the cited publications demonstrated the beneficial effects of both PIR and MR methods in reducing pain sensations and improving masticatory organ function. However, it should be noted that Ibanez-Garcia et al. [17] demonstrated a significantly greater improvement in the pain-free mandibular range of motion after using the modified MR method than using the PIR intervention described by Blanco et al. [16]. This finding, reflected also in the results of this study indicates the possible greater efficiency of myofascial relaxation in improving the functioning of the masticatory system. Additionally, Cuccia et al. [45] and Raiadurai [46] investigated the use of PIR [45,46] and MR methods [45] in TMD patients. Both authors assessed the clinical effectiveness of treatment protocols consisting of several therapeutic techniques. They reported an improvement of clinical outcomes—however, it is clearly not possible to selectively assess the impact of each of the discussed methods on the clinical condition of treated patients.

More data on the effects of PIR and MR methods can be found in reports describing the use of these techniques in trunk and limbs soft-tissue therapy [27,32,33,34,35,36,37,38]. These reports indicate a significant effect of PIR on the minimization of pain and increasing the range of motion in the joints [16,27,32,33,34], accompanied by a decrease in muscle electrical activity [35]. Although a wide variety of MR techniques can be found in the literature [47,48,49,50,51,52,53,54,55]. Authors assessing the outcomes of MR procedures, methodically consistent with the approach carried out in this study, indicate their effectiveness in increasing the pain-free range of motion [17], reducing pain [17,36,37], improving musculoskeletal function [36,37] and reducing muscle stiffness [38]. In the reports that have shown a direct relationship between the level of perceived pain and muscle electrical activity [56,57], all of the aforementioned findings are consistent with the results of the present study.

Summarizing the above considerations, it should be noted that the results of this research are confirmed by current scientific reports and indicate the positive effects of using both PIR and MR methods in the supportive treatment of TMD. Both methods reduce pain and tension in the masticatory muscles.

### Study Limitations

One of the main limitations of the study seems to be the small sample size. In order to confirm the results, it would be worthwhile to conduct a multicentre study that would allow for an analysis of a larger sample and minimize the data collection time. Although strenuous efforts were made and anthropometric points were used as well as photographic documentation of the sEMG study to ensure the repeatability of the electrode locations, it is difficult to say with certainty that the electrode application points were exactly the same.

It is also worth noting that a comparison of the results of the present study with the results described by other researchers requires a comparison of the technical specifications for the measuring devices used for electromyographic assessment. A direct comparison of the values of muscle electrical activity may not be possible due to the different technical parameters of measuring devices, such as the sensitivity or frequency of measurement.

## 5. Conclusions

The use of both post-isometric muscle relaxation and myofascial release methods reduces the increased tension of the anterior part of the temporal and masseter muscles in the group of patients treated due to a temporomandibular disorder.The use of post-isometric muscle relaxation and myofascial release methods reduces the electrical activity of the masticatory muscles in the rest position of the mandible to a similar degree.The use of post-isometric muscle relaxation and myofascial release methods reduces the intensity of spontaneous pain in the masticatory muscles to a similar degree.

The results of this study indicate that both the PIR and MR methods can be used as effective forms of supportive therapy in the prosthetic treatment of pain-related TMD, accompanied by increased masticatory muscle tension.

## Figures and Tables

**Figure 1 ijerph-18-12970-f001:**
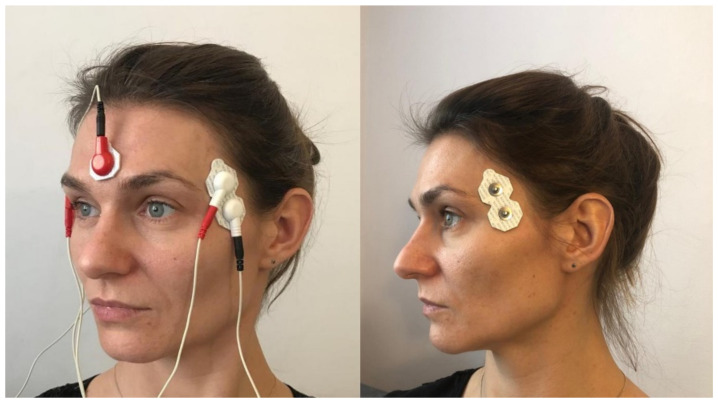
Location of surface electrodes on the front of the temporal muscle. (Own material).

**Figure 2 ijerph-18-12970-f002:**
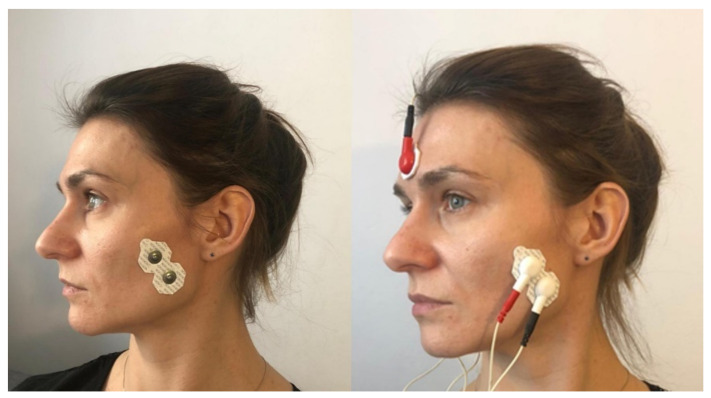
Location of surface electrodes on the masseter muscle.

**Figure 3 ijerph-18-12970-f003:**
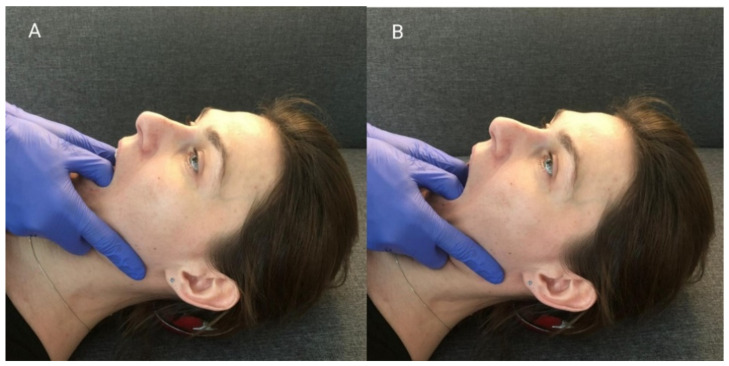
PIR procedure for mandibular adductors: (**A**) Starting position; (**B**) passive abduction of the mandible.

**Figure 4 ijerph-18-12970-f004:**
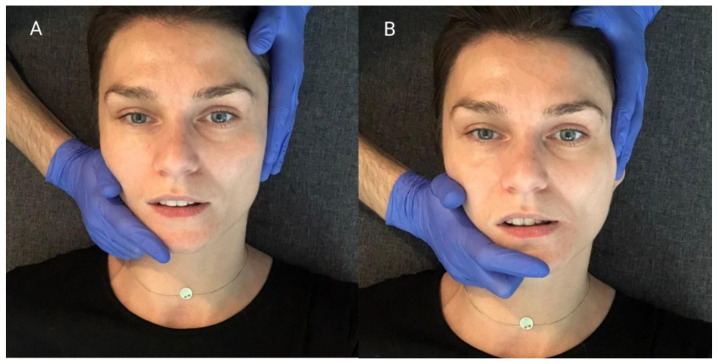
PIR treatment of the muscles responsible for the lateral movement of the mandible towards the right: (**A**) Starting position; (**B**) passive lateral movement of the mandible.

**Figure 5 ijerph-18-12970-f005:**
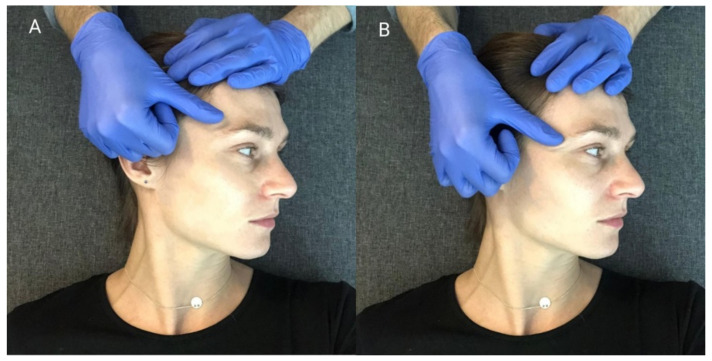
MR procedure of the muscle on the right side: (**A**) Locating the restrictions; (**B**) taking out the tissue slack.

**Figure 6 ijerph-18-12970-f006:**
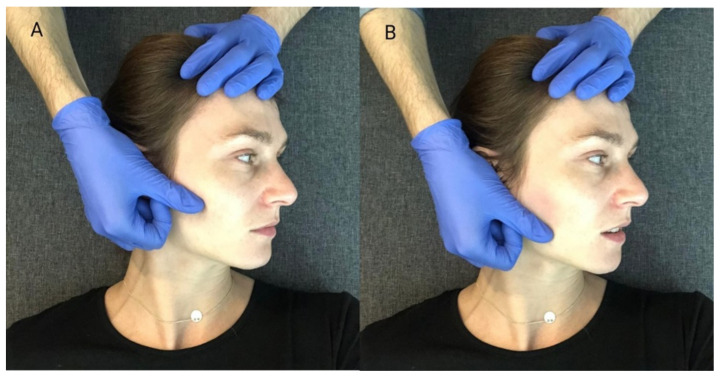
MR procedure of the masseter muscle on the right side: (**A**) tacking out the tissue slack; (**B**) Mobilization of the myofascial complex.

**Figure 7 ijerph-18-12970-f007:**
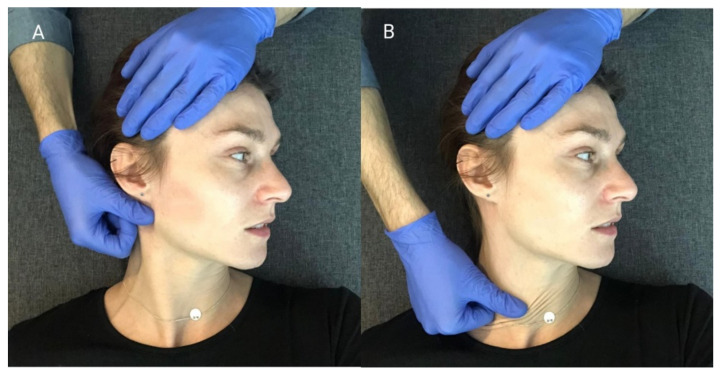
MR procedure of the sternocleidomastoid muscle on the right side: (**A**) taking out the tissue slack; (**B**) mobilization of the myofascial complex.

**Figure 8 ijerph-18-12970-f008:**
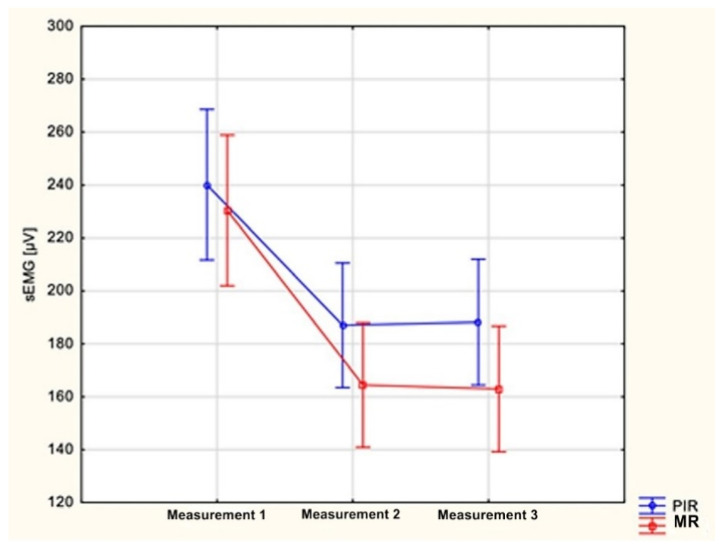
Comparison of changes in the mean values of electric potentials [µV] of left masseter muscles measured during electromyographic examinations between group I (PIR) and group II (MR); *p* = 0.11.

**Table 1 ijerph-18-12970-t001:** Comparison of the distribution of the sex, age, mean values of electric potentials [µV] of the anterior part of temporal muscles and masseter muscles, and the intensity of spontaneous pain in the masticatory muscles in the VAS score assessed during the first examination between group I and group II.

	Group I			Group II		*p*
		**Sex**				
Woman	22			25		0.35 ^A^
Man	8			5		
		**Age**				
Years	Mean	SD		Mean	SD	
	28.37	5.31		27.77	5.10	0.66 ^B^
		**Temporal muscles**				
sEMG [µV]	Mean	SD		Mean	SD	
Right	251.5	83.92		243.0	83.18	0.70 ^B^
Left	249.9	82.49		241.3	80.21	0.68 ^B^
		**Masseter muscles**				
sEMG [µV]	Mean	SD		Mean	SD	
Right	239.4	76.47		228.8	73.67	0.59 ^B^
Left	240.1	77.42		230.4	78.40	0.63 ^B^
			**Pain**			
	Median	q1–q3		Median	q1–q3	
	5.0	4–7		4.5	3–6	0.53 ^C^

q1—the first quartile, q3—the third quartile; ^A^—chi-square test; ^B^—*t*-test; ^C^—*Mann–Whitney* U test; sEMG—surface electromyography.

**Table 2 ijerph-18-12970-t002:** Results of the physiotherapeutic examination of the masticatory organ obtained in group I.

Symptoms		Subject Number		
		Examination 1	Examination 2	Examination 3
The range of active depression of the mandible	increased	-	-	-
	decreased	6	-	-
Deviations from the sagittal plane during active depression and elevation of the mandible		10	5	5
The range of active lateral movements of the mandible	increased	-	-	-
	decreased	5	-	-
Pain in the masticatory muscles during palpation (VAS greater or equal to 5)		29	17	17
Pain in the temporomandibular joints during palpation (VAS greater or equal to 5)		21	10	11
Worsening of pain in joints, periarticular structures or muscles during passive mandible movements		24	17	18

VAS—visual analogue scale.

**Table 3 ijerph-18-12970-t003:** Results of the physiotherapeutic examination of the masticatory organ obtained in group II.

Symptoms		Subject Number		
		Examination 1	Examination 2	Examination 3
The range of active depression of the mandible	increased	-	-	-
	decreased	3	-	-
Deviations from the sagittal plane during active depression and elevation of the mandible		7	4	5
The range of active lateral movements of the mandible	increased	-	-	-
	decreased	1	-	-
Pain in the masticatory muscles during palpation (VAS greater or equal to 5)		27	12	13
Pain in the temporomandibular joints during palpation (VAS greater or equal to 5)		13	6	6
Worsening of pain in joints, periarticular structures or muscles during passive mandible movements		28	21	19

VAS—visual analogue scale.

**Table 4 ijerph-18-12970-t004:** Mean values of electric potentials [µV] of the anterior part of temporal muscles and masseter muscles measured during the first examination and control tests in group I.

	Temporal Muscles				Masseter Muscles			
	Right		Left		Right		Left	
**sEMG [** **µ** **V]**	**Mean**	**SD**	**Mean**	**SD**	**Mean**	**SD**	**Mean**	**SD**
Measurement 1	251.5	83.92	249.9	82.49	239.4	76.47	240.1	77.42
Measurement 2	186.6	65.58	187	66.12	187.4	62.21	187.0	62.22
Measurement 3	190.3	68.92	189.1	67.31	188.6	65.01	188.2	65.61
*p*	<0.001 ^A^		<0.001 ^A^		<0.001 ^A^		<0.001 ^A^	

sEMG—surface electromyography; ^A^—ANOVA test for repeated measures.

**Table 5 ijerph-18-12970-t005:** Mean values of electric potentials [µV] of the anterior part of temporal muscles and masseter muscles measured during the first examination and control tests in group II.

	Temporal Muscles				Masseter Muscles			
	Right		Left		Right		Left	
**sEMG [** **µ** **V]**	**Mean**	**SD**	**Mean**	**SD**	**Mean**	**SD**	**Mean**	**SD**
Measurement 1	243.0	83.18	241.3	80.21	228.8	73.67	230.4	78.40
Measurement 2	170.9	70.45	169.5	66.54	162.5	63.08	164.5	66.49
Measurement 3	169.7	67.17	168.0	63.49	161.2	61.19	162.9	64.46
*p*	<0.001 ^A^		<0.001 ^A^		<0.001 ^A^		<0.001^A^	

sEMG—surface electromyography; ^A^—ANOVA test for repeated measures.

**Table 6 ijerph-18-12970-t006:** Intensity of spontaneous pain in the masticatory muscles in the VAS score, assessed during the first examination and follow-up in group I (PIR) and group II (MR).

	PIR		MR	
Pain	Median	q1–q3	Median	q1–q3
Measurement 1	5.0	4–7	4.5	3–6
Measurement 2	2.0	2–3	2.0	1–3
Measurement 3	2.0	2–3	2.0	2–3
*p*	<0.001 ^A^		<0.001 ^A^	

PIR—post-isometric relaxation, MR—myofascial release; q1—the first quartile, q3—the third quartile; ^A^—Friedmann test.

**Table 7 ijerph-18-12970-t007:** Comparison of changes in the intensity of spontaneous pain in the masticatory muscles (Pain) for the VAS score assessed during the first study and follow-up between group I (PIR) and group II (MR).

	PIR		MR		
Pain	Median	q1–q3	Median	q1–q3	*p*
Change between measurement 1 and measurement 2	2	2–4	2	1–4	0.77 ^A^
Change between measurement 1 and measurement 3	2	1–4	2	1–3	0.90 ^A^
Change between measurement 2 and measurement 3	0	0–0	0	0–0	0.45 ^A^

PIR—post-isometric relaxation, MR—myofascial release; q1—the first quartile, q3—the third quartile; ^A^—Wilcoxon test.

## Data Availability

The data presented in this study are available on request from the corresponding author. The data are not publicly available due to privacy restrictions.

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
