# Peer review of "The Application of Manual Techniques in Masticatory Muscles Relaxation as Adjunctive Therapy in the Treatment of Temporomandibular Joint Disorders"

_ijerph, 2021, doi:10.3390/ijerph182412970_

Round 1

Reviewer 1 Report

This paper aims to compare the degree of relaxation of the anterior part of the temporal muscles and the masseter muscles using two different methods of physical therapy (post-isometric relaxation- PIR; myofascial release – MR). This comparison was carried out using a surface EMG instrument, tool reported by the authors as applied for the first time for such comparison.

Despite the topic, that could be interest, the manuscript is not clear and not presented in a well-structured manner, and the obtained results don’t provide an advance towards the current knowledge. The language needs to be entirely revised, resulting mainly not appropriate and with several passages not understandable in the manuscript.

The entire paper, including the introduction and discussion sections should be summarized and reorganized in a more well-structured way. The study design sounds correct, but is not evaluable in a correct way since the methodology is not clear, criteria of inclusion in the study are not sustained by disease classifications, 

Author Response

Dear Sir / Madam,

Kind regards,

Piotr Urbański

Reviewer 2 Report

The paper itself is well written and documented, showing a great effort from the authors. The topic sounds original and with an interesting clinical meaning and the large sample size is a point of strength.

I only have some suggestions:

  • DC/TMD should be added in the section describing inclusion criteria
  • a brief description of the exactly location of the electrodes for sEMG could be added
  • a random distribution of the subjects in the two groups should be adopted in this type of prospective study preferring the RCT study design
  • consider the possibility to cite these papers recently published regarding TMD epidemiology and treatment DOI:10.3390/ijerph17238760 ; DOI:10.3390/app11188303

Otherwise the article is good.

Author Response

(The authors gave the same response as above.)

Round 2

Reviewer 1 Report

Abstract

I group and II, I would call them group I and group II, otherwise there is implied a meaning of different relevance between the two groups (temporal, treatment etc). Furthermore, they are called group I and group II later. The use of the same terminology is needed.

“the comparative assessment…(VAS)”. In this sentence is not clear the part concerning the pain. It’s better delete “the data on”, leaving just “the intensity…”

Keywords. TMD is already present as keyword, both in the extensive and short name, so “treatment of TMD” can be deleted.

Materials and methods

The trial was not prospectively registered, but this is an issue for the editor to decide.

Patients seeking for prosthetic treatment were recruited, but the inclusion criteria (v) are the requirement of full dental arches. This point needs to be clarified. (this point is also addressed by the authors at the end of the conclusion section)

The authors refer to the RDC/TMD classification but at the time of the recruitment of patients (theorically 2016, based on the ethical approvement, since there are no temporal indications of the study) this classification had already been replaced by the DC/TMD classification. This last classification is mentioned later, so what kind of TMD classification was used for this study?

This schedule was required to conduct diagnostic and therapeutic meetings in the time required for the laboratory implementation of occlusive splints, used as primary TMD therapy. That procedure also enabled manual therapy to be carried out as the first treatment and thus avoid the influence of confounding factors like other medical interventions.” What does this paragraph mean? It needs to be clarified

Physical examination, once is called in this way and then is called Physiotherapeutic examination. The authors need to use and repeat the same terms for clarity, in general in the entire paper.

Results

“Before the therapy there was no difference between the groups in the distribution of sex, age, electromyographic tension of the anterior part of the temporal and masseter muscles, and the intensity of spontaneous pain in the masticatory muscles” This sentence need to be sustained from a table with these basic data analysed in the sample and the proper statistic test used for the comparison.

The paragraph 3.2 is not clear. All the results need to be reported entirely in the referring tables (3-4). What does the percentage values exposed in the text refer to? The anova test was used for comparing the data during the three examinations, while which was the test used for the paired analysis? (ex1-2 and ex2-3)

The results related to paragraph 3.3 are not statistically significant, so figure 8-11 are not necessary, or at least they could be included in one figure only. This was also addressed before. Table 5 and 6 are not clear, why there are no data concerning the mean values of spontaneous pain? “q1-q3” is referring to? This was also addressed before.

In all Tables the specific statistical test used is still missing, as addressed before.

The reported references are too many, mainly dated before year 2016, and include an abnormal number of self-citations. This is another issue that still remain and was not modified.

Author Response

Dear Sir / Madam,

Your faithfully,

Piotr Urbański
